# Polysiloxane Coatings Biodeterioration in Nature and Laboratory

**DOI:** 10.3390/microorganisms10081597

**Published:** 2022-08-08

**Authors:** Maxim Danilaev, Galina Yakovleva, Sergey Karandashov, Vladimir Kuklin, Hong Quan Le, William Kurdy, Olga Ilinskaya

**Affiliations:** 1Department of Electronic and Quantum Means of Information Transmission, Kazan National Research Technical University N.A. A.N. Tupolev-KAI, 420111 Kazan, Russia; 2Microbiology Department, Kazan Volga-Region Federal University, 420008 Kazan, Russia; 3Seaside Branch of Russian-Vietnamese Research and Technology Center, Nha Trang 625080, Vietnam

**Keywords:** organic glasses, polysiloxane coating, bio-damage, tropics, micromycetes, abrasion, adhesion, transparency

## Abstract

Objects and structures made of organic glass require protection from damage caused by external factors. Light, humidity, temperature, dust pollution and, undoubtedly, microorganisms lead to the deterioration of optical and mechanical properties. Polysiloxane-based protective coatings, consisting of silicon–oxygen backbones linked together with organic side groups attached to the silicon atoms, are widely used. However, the polysiloxane coatings themselves also cannot avoid deterioration during operation that implies the constant development of new protective materials. Here, we created a new cross-linked polysiloxane that covers organic glasses to enhance their resistance to aggressive external factors, and investigated its own resistance to damage induced by micromycetes in natural tropical conditions and in the laboratory. It has been established that the surface of coatings in the tropics is prone to fouling with micromycetes, mainly of the genera *Aspergillus* and *Penicillium*, which produce oxalic, malic, lactic, and citric acids contributing to the biodeterioration of polysiloxane. The testing of monolithic polycarbonate, polymethyl methacrylate, and triplex coated with polysiloxane showed that they retained significant resistance to abrasion and transparency at a level of more than 90% under aggressive natural conditions. Under artificial laboratory conditions, the infection of samples with micromycete spores also revealed their growth on surfaces and a similar trend of damage.

## 1. Introduction

Various organic glass materials, namely polycarbonate and polymethyl methacrylate, are used for glazing vehicles, buildings, and structures. The application of polysiloxane coatings is one of the well-known methods of protecting these materials from aggressive external factors leading to abrasion and a decrease in the transparency. By varying the -Si-O- chain length, side groups, and crosslinking extent, the silicones with properties ranging from liquids to hard plastics can be synthesized [1]. However, the polysiloxane coatings themselves also cannot avoid aging, which reduces the service life of the polymer materials. In the environment, especially in tropical climates, the coatings are exposed not only to ultraviolet radiation, high humidity, and high temperatures, but also to microorganisms. Unlike medical materials, where the formation of bacterial biofilms plays a key role in coating degradation [2,3,4,5], industrial polymeric materials lose more of their optical and mechanical properties under the action of microscopic fungi, the spores of which settle on the surface and are able to germinate.

The existing methods of preventing the degradation of polysiloxanes or reducing its rate are based on the formation of a composition with additional components. So, to increase the resistance to UV exposure, compositions with substances that transform the UV radiation spectrum into the visible range are used [6,7]. An increase in the resistance to high temperatures can be achieved by forming a cross-linked structure of a polysiloxane film using Si-H side groups and eugenol [8]. The known methods for reducing the rate of biodegradation are based on the use of dispersed particles, for example, silver nanoparticles, copper dioxide, or the introduction of functional groups into the compositions that increase the toxicity of the materials to microorganisms [9,10,11].

All of the methods used to improve the resistance of the polysiloxane coatings to external influences have several fundamental limitations. For example, the introduction of nanoparticles that exhibit toxicity to microorganisms [12,13] or transform the spectrum of UV radiation [6,7] leads to a decrease in the optical characteristics of the glazing, although the particle sizes are smaller than the wavelengths of the optical range [14,15]. Strengthening the antimicrobial properties of the polysiloxane coatings due to functional groups’ modification leads to a decrease in the abrasion resistance [16]. Today, the search for ways to create a polysiloxane composition that is resistant to external destructive influences (UV radiation, high temperature, humidity, microorganisms) and, at the same time, retains its performance characteristics is an important task. In the present work, we have created a polysiloxane coating for organic glasses intended for use in tropical climates, and have analyzed its stability in natural and laboratory conditions, paying special attention to the destructive role of micromycetes. Comprehensive analyses of the micromycetes associated with the deterioration of the polysiloxane coating may contribute to the understanding of mechanisms of deterioration, as well as to the identification of potentially undesirable fungal communities or extremely aggressive species.

First, we have set an aim to isolate and identify the microscopic fungi from polysiloxane-coated samples placed for seven months in the natural conditions of a tropical climate in Vietnam. Second, we analyzed the aggressive factors of these fungi and the possible stages of destructive mechanisms. Third, model tests of the studied polysiloxane coatings for fungi resistance in laboratory were carried out. Finally, we characterized the fungi-induced damage of polysiloxanes in terms of optical density, abrasion resistance, and adhesion strength of polysiloxane coatings to organic glasses.

## 2. Materials and Methods

### 2.1. Polysiloxane Coating

The varnish from which the polysiloxane coating was formed was obtained by the reaction of hydrolysis of methyltrimethoxysilane (Dow Corning Europe SA, Seneffe, Belgium) in an acidic medium at pH 3, as described earlier [17]. A silica dispersion (Dynasylan^®^ SIVO 140, manufactured by Evonik Corporation, Theodore, AL, USA) weighing 25 ± 0.5 g was loaded into a 300 mL round bottom flask and rotated on a magnetic stirrer at a speed of about 600 rpm. The catalyst (acetic acid, 96%, Merck Life Science LLC, Darmstadt, Germany) was then loaded into this flask and stirred for 15 min. The mass of the catalyst was 0.01 ± 0.002 g. Then, the distilled water was added in a volume of 15 mL and stirred for 15 min. After that, 25 mL of a mixture of isopropanol:butanol (1:1) was added (mixture of isopropanol/butanol = 1/1; 2-Propanol, CAS-No. 67-63-0; Butyl alcohol 99.9%, CAS-No. 71-36-3, manufactured by Merck Life Science LLC Darmstadt, Germany) and mixed for 15 min.

Then, 10 g of monomer (methyltriethoxysilane OFS-6070, CAS: 1185-55-3, manufactured by Dow Corning Europe S.A.) was added at a rate of five drops/min, and the solution was stirred for 24 h. To impart adhesive and hydrophobic properties to the varnish, the vinyl groups (vinyltrimethoxysilane, Evonik Degussa GmbH, Essen, Germany) and the octyl groups (octyltriethoxysilane, XIAMETER OFS-6341, Dow Chemical Company, Kankakee, IL, USA) were introduced, respectively, in an amount of 10% of dry weight. The dry weight of the varnish was determined experimentally by weighing the control sample of the varnish after it had dried. The drying was carried out at a temperature of 150 ± 5 °C for 3 h. Weighing was performed on an analytical balance, with an error of no more than ±0.1 mg.

To reduce the evaporation rate of the solution mixture, propylene glycol was added, which gave it a well-cross-linked structure.

### 2.2. Organic Glass Samples

Three types of organic glass samples of 50 mm × 50 mm were used: (1)—monolithic polycarbonate; (2)—monolithic polymethyl methacrylate; (3)—polycarbonate 4 mm thick glued to polycarbonate 8 mm thick (triplex). The cleaned and degreased samples (20 pieces of each type) were dipped into the obtained varnish with a dry weight of 26%. The rate of lifting samples from the container with the varnish was ~5 mm/s. To form a polymer film, the samples were placed in an oven, heated to 150 °C and held for ~3 h, followed by natural cooling to room temperature (22 ± 3 °C). The coating thickness was measured with a DektakXT^®^ probe profilometer with a standard error of 10 nm. The coating thickness of all of the samples was 20 ± 4 μm.

The scheme of the varnish composition formation on the organic glass surface is shown in Figure 1.

### 2.3. Natural Trials of Polysiloxane Coatings in the Tropics

The organic glass samples coated with polysiloxane (10 samples of each type one, two, and three) were kept for 12 months at the climatic testing stations of the joint Russian–Vietnamese Research and Technology Center (hereinafter referred to as the Tropical Center) (Figure 2). After the exposure, the samples were delivered to the laboratory, where the identification of the strains in the detected microbial fouling and analysis of the changes in the mechanical and optical properties of the samples were carried out.

### 2.4. Isolation and Taxonomic Identification of Micromycetes

From the samples on which the microbial fouling was recorded, a wash was completed with sterile water and sown onto Petri dishes with Czapek-Dox medium.

After 7 days of cultivation at 30 °C, the individual colonies of micromycetes were microscopically examined for preliminary identification, according to the morphological features. Additionally, we analyzed the sequences of ITS (Internal Transcribed Spacer), a highly polymorphic non-coding region of nuclear DNA (rDNA) which is the most sequenced region to identify fungal taxonomy at species level [18]. Some of the primers, e.g., ITS1-F, ITS1, and ITS5, are biased towards the amplification of basidiomycetes, whereas others, e.g., ITS2, ITS3, and ITS4, are biased towards the ascomycetes which are amplified more easily than the basidiomycetes using these ITS as targets [19]. The polymerase chain reaction (PCR) amplification of the ITS region from each of the tropics-exposed isolates was performed using ITS-1 as a forward primer, and ITS-4 as a reverse primer [20,21].

The mycelium fragments were crushed by vigorous vortexing with 5 mm glass beads (Sigma, West Lafayette, IN, USA). The DNA was isolated using a commercial Fast DNA spin kit for soil (MP Biomedicals), according to the manufacturer’s instructions. The concentration of the isolated DNA was measured using a Qubit fluorometer (Invitrogen, Carlsbad, CA, USA).

The PCR was performed on a 50 μL mixture containing 0.2 mM dNTP (SibEnzyme, Novosibirsk, Russia), 20 pmol of each primer (Sintol, Moscow, Russia), 2.5 u.a. Taq polymerase (Sintol, Moscow, Russia), 2.5 mM MgCl2 (Sintol, Moscow, Russia), and 2 ng of genomic DNA. Amplification protocol: initial denaturation 3 min—95 °C; 30 cycles (30 s—95 °C, 30 s—50 °C, 1 min 10 s—72 °C); final elongation 10 min—72 °C. The resulting fragments were separated by electrophoresis (1.5% agarose gel in Tris-borate-EDTA buffer with ethidium bromide) and visualized in UV light. The PCR products were purified using an Omnix DNA purification kit (Omnix, St. Petersburg, Russia).

The DNA sequencing was performed, according to the Sanger method, on an ABI Prizm 3500 genetic analyzer (Applied Biosystems, San Francisco, CA, USA). The reaction was carried out using the BigDye Terminator v3.1 Cycle Sequencing Kit (Applied Biosystems, San Francisco, CA, USA) in a volume of 10 µL. The purification of terminating nucleotide analogs was completed using the BigDye XTerminator Purification Kit (Applied Biosystems, San Francisco, CA, USA). The sequencing was performed in both directions. Identification was carried out in accordance with the criteria of CLSI [22]. All of the inconsistencies between the phenotypic and molecular genetic identifications were verified by resequencing.

### 2.5. Laboratory Trials of Polysiloxane Coatings

The laboratory trials of coating resistance were carried out using three fungal species: *Aspergillus niger*; *A. puulaauensis*; and *Penicillium chrysogenum.* The samples were infected with fungal spores in the absence of mineral and organic substrates (option 1), and under conditions imitating the mineral and organic contaminants (option 2). The spores of three dominant fungal strains isolated from polysiloxane fouling in the Tropical Center were used at a concentration of 4 × 10^6^ per mL of distilled water for each micromycete. All of the sides of the samples were evenly sprayed with the mixed suspension of the spores or sterile distilled water as a negative control (option 1), or were placed in 100 mL of Czapek-Dox medium containing 1 mL of mixed spore suspension (option 2). The scheme of the experiments is shown on Figure 3. The samples were kept in chambers at 30 °C and a relative air humidity of more than 90% during 21 days, with an intermediate examination every 7 days.

### 2.6. Organic Acids Analysis

A cultural fluid of fungal isolates growing for 5 days up to the end of the exponential phase on liquid Czapek-Dox medium was analyzed for the production of secreted organic acids, using a high-performance liquid chromatograph Flexar (PerkinElmer, Waltham, MA, USA). The separation of the organic acids was carried out in a Brownlee Analytical C18 (150 × 4.6 mm) column packed with 3 µm particles. The elution was carried out at room temperature in a linear gradient, using a system consisting of a solution of 10 mM KH_2_PO_4_ adjusted with orthophosphoric acid to pH 2.4 (eluent A) and acetonitrile (eluent B). The flow rate of the mobile phase was 1 mL/min. The peaks were identified using a UV detector at 210 nm (Reuter 2015). As the standards, acid solutions were used (g/L): oxalic (0.15); tartaric (0.35); malic (0.5); lactic (0.35); citric (0.5); and acetic (0.15).

### 2.7. Optical Properties Control

The transmittance (τ) and turbidity (H) of the samples were determined on a pulse photometer, constructed and described by [23]. A block diagram of the photometer is shown in Figure A1. The photometer was covered with an opaque casing to prevent the influence of external light sources on the measurement results. Multiple measurements of the control clean glass slides with known optical characteristics (Table A1) showed that the instrumental standard root-mean-square deviation of the transmission coefficient measurement results did not exceed 1.4%, and the same value of the sample turbidity measurement results did not exceed 2.7% [14]. The transmittance of the samples was determined as the ratio of optical radiation intensity transmitted through the samples to the incoming irradiation intensity. The intensities were measured using a photodetector (photodiode FD256 with a spectral sensitivity range of 0.4–1.1 µm). A light-emitting diode BL-L101URC with a wavelength of optical radiation of 660 nm was used as a light source. The samples’ turbidity relates to the scattering of radiation due to the polysiloxane coating’s destruction. The turbidity was determined by comparing the intensity of the scattered optical radiation of the experimental and control (non-treated) samples.

### 2.8. Abrasion Measurement

The polysiloxane-coated sample was fixed onto the surface of a rotating disk (Ø = 136 mm). The tip (Ø = 5 mm), wrapped in an abrasive cambric cloth, was pressed onto the sample’s surface at a distance of 5 mm from the disk rotation axis. The disk with the fixed sample was rotated (speed = 40 rotations/s, the pressing force ≈ 2 N, the sliding speed 0.3 m/s). After 60 s, the sample was examined, using a Levenhuk 320 microscope (Levenhuk Inc., Tampa, FL, USA), and the abrasion of the coating was determined by the presence or absence of scratches on its surface.

### 2.9. Adhesion Measurement

A cross-cut test (ISO 2409:2013) was used to assess the adhesion of the polysiloxane coating to the surface of organic glasses. The test was carried out at 23 °C and relative humidity of 50%. The polysiloxane coating (thickness of ~20 μm) was damaged through the entire thickness by cuts (length of cut = 20 mm, their number = 20) applied as a grid with a step of 1 mm. The transverse incisions were made manually using a knife (ISO 2409:2013) with a V-shaped blade: blade thickness 0.43 ± 0.03 mm; sharpening angle 25 ± 3°; sharp edge width ~0.05 mm. A transparent polyethylene terephthalate (PET) tape was attached to the cut areas with rubber glue and abruptly detached. Then, the edges of the cuts were analyzed using the microscope, Levenhuk 320 (Levenhuk Inc., Tampa, FL, USA), according to ISO 2409–2013.

## 3. Results

### 3.1. Characteristics of Isolated Fungi and Their Destructive Properties towards Polysiloxane-Coated Organic Glasses

#### 3.1.1. Fungal Isolates from the Samples’ Surface after Exposure in Tropical Conditions

The analysis of the samples exposed for 12 months in the Tropical Center revealed a slight fouling of the surfaces of all of the samples. Sample 3 was visually more prone to fouling than the other two samples. The pure cultures of the micromycetes were isolated from fouling by classical cultivation methods on solid Czapek-Dox medium. Based on microscopic and molecular genetic analyses, the isolates were assigned to the following species: *Aspergillus niger*; *Aspergillus puulaauensis; Aspergillus fumigatus; Penicillium chrysogenum; Penicillium griseofulvum; Fusarium graminearum;* and *Alternaria alternate*. The genus *Aspergillus* was represented by the maximal colony numbers isolated from each sample, with a clear dominance of the *A. niger* species. The raw sequence data produced in this study were deposited in NCBI, SRA (Sequence Read Archive) under the accession No SRR18883168 (https://www.ncbi.nlm.nih.gov/sra/SRR18883168 (accessed on 7 May 2022).

#### 3.1.2. Fungal Resistance Testing in Laboratory Conditions

Dominant micromycetes of three species *A. niger*, *A. puulaauensis*, and *P. chrysogenum* were selected for laboratory testing. Septate hyphae of the germinated spores of the mycelial fungi, single linear ones (sample 1), and intertwined or branched ones (samples 2 and 3), were found on the surface of all of the samples. The characteristics of the two types of sporangiophores indicate the growth of *Aspergillus* and *Penicillium* on the polysiloxane coating. Fundamental differences in the fouling of the samples treated according to option 1 and option 2 were not found, except for the earlier spore germination after 7 days when using option 2, compared with 14 days when using option 1. Although option 2 allows for the intensive germination of spores on the Czapek-Dox medium containing carbon and nitrogen sources, the fouling of the coatings was essentially the same as for option 1. The morphological characteristics of sporangiophores indicate the predominant growth of *A. niger*, represented by a branched septate mycelium (Figure 4).

#### 3.1.3. Organic Acids Synthesized by Dominant Micromycetes Isolated in Tropics

The main secreted metabolic-products of the micromycetes that contribute to biocorrosion are carboxylic acids, leading to a significant decrease in pH. We have analyzed the spectrum of the organic acids synthesized by *A. niger, P. chrysogenum*, and *A. puulaauensis*. All of the studied fungi were capable of synthesizing oxalic, malic, and citric acids (Figure 5). The highest amount of oxalic acid (410.14 ± 12.30 mg/L) was detected in the culture liquid of *A. puulaauensis*; malic (441.73 ± 39.76 mg/L) and citric (77.74 ± 5.44 mg/L) acids were produced at a maximal level by *A. niger*. The lactic acid was synthesized by *A. niger* and *A. puulaauensis*. Acetic acid in a small amount (0.77 ± 0.07 mg/L) was found in the culture liquid of *P. chrysogenum*. Tartaric acid was not found anywhere. Large amounts of oxalic acid were detected in all of the fungi, the dominant strain *A. niger* also synthesized a high amount of malic acid and was the most active producer of organic acids.

#### 3.1.4. Changes in the Optical–Mechanical Properties of Polysiloxane Coatings

The results of the experimental studies indicate the effect of fungi on the optical characteristics of all of the samples. Taking the transmittance of the non-treated control sample as 100%, it can be calculated that the transmittance decreases most significantly for the samples exposed in the tropics, by 32.1%, 27.6%, and 29.1% for samples 1, 2, and 3, respectively (Table 1). A less significant decrease in the transmittance was recorded in laboratory conditions, while the samples treated according to option 2 showed a greater decrease in transmittance than those treated according to option 1. The polysiloxane-coated monolithic polymethyl methacrylate (sample 2) retained its optical characteristics better than the other two samples, also including the smallest change in turbidity. Compared to the control samples, the turbidity of samples 1, 2, and 3 under the tropical conditions increased by 39.5, 9.3, and 9.8 times, respectively (Table 1). However, large values of these calculated data are actually of relative importance, and are given for a clearer comparison of the damage degrees of the three samples. In fact, if the complete absence of the optical radiation transmission was taken for 100% of turbidity, the turbidity of the samples over the entire exposure time in the tropics did not exceed 10% (Figure 6). The mechanical properties of the samples changed insignificantly. A slight abrasion of the polysiloxane coating was detected after exposure in the tropics for samples 2 and 3, while sample 1 was also subjected to abrasion after the option 2 treatment.

After the tropical trials, no change in the adhesion strength of the polysiloxane coating on the surface of the organic glasses was observed. There was a slight deterioration in the adhesion strength of the coating for sample 1. However, such deterioration was observed only for a part of sample 1 (one of ten samples) treated according to option 2 (Figure 7).

## 4. Discussion

One of the important requirements for industrial and construction facilities operating in contact with an aggressive environment is their long-term resistance to bio-damage. For optically transparent organic materials, there is a constant search for new coatings that ensure the preservation of the optical–mechanical properties and, at the same time, are resistant to biofouling. The polysiloxane synthesized in this work is practically a cross-linked structure that covers organic glasses to enhance their resistance to biodeterioration. This network lacks heteroatoms, for example, P, B, Ti, Al, the inclusion of which makes polysiloxanes more susceptible to thermal degradation [24]. Thus, the polysiloxane varnish used in the work has good protective properties, which is confirmed by the transparency preservation of all of the samples at a level of more than 90% under the most aggressive conditions of the tropics (Figure 5). The base component, namely organic glass on which the varnish is applied, affects the optical characteristics of the samples under fungal influence. So, sample 1 (monolithic polycarbonate) turned out to be the most susceptible to the loss of optical characteristics, while sample 2 (monolithic polymethyl methacrylate) and sample 3 (polycarbonate triplex) were approximately the same in terms of stability (Table 1). Although the test period in the tropics was 48 weeks, and only 3 weeks in the laboratory, the results of the experiments are consistent. Sample 1 was the least stable, showing a 15.5-fold increase in turbidity compared to the untreated samples; samples 2 and 3 were equally stable. As expected, in the presence of the additional nutrients for fungal growth in the Czapek-Dox medium (option 2), biodeterioration was more prominent compared to option 1 without the added nutrients. However, sample 1 again showed the largest increase (14-fold) in turbidity (Table 1). No significant changes in the adhesion strength of the polysiloxane coating on the surface of the organic glasses were observed for all of the three options.

The service life of polysiloxane can be 10 years at 180 °C. However, under the light, photochemical transformations occur in the polymer. The chromophore groups containing multiple bonds or an atom with a free pair of electrons pass upon absorption of photon into an activated state, the energy of which can exceed the dissociation energy of the chemical bond. In this case, the bond dissociates with the formation of radicals, which causes secondary, so-called “dark”, decomposition reactions. The polymers containing the chromophore groups that absorb light in a region close to the spectrum of sunlight near the Earth’s surface (λ ≥ 270 nm), namely, carbonyl (λ = 279; 285 nm) and aromatic (λ ≥ 193; 260 nm) undergo intense photo-destruction. However, almost all of the polymers undergo photoaging under natural light. This is due to the content in the chromophore groups of random impurities (plasticizers, stabilizers) in the composition and in the oxidation products of the polymer. Although the coating we used does not contain carbonyl and aromatic chromophore groups (Figure 1), we cannot completely exclude photo-destructive processes occurring in tropical climates. At the same time, we have shown that, in the absence of them in laboratory tests, polysiloxane also loses its optic quality. This indicates the predominant fungal contribution to the deterioration of polysiloxane under high humidity, high temperature, and the presence of additional nutrient sources for micromycetes. The temperature in the laboratory tests was 30 °C at 90% humidity; in the tropics, the temperatures ranged from 26 °C to 38 °C (average temperature 26.7 °C), with a humidity of about 81%. These conditions are similar; however, the incubation time of samples in tropics and photo-destruction may contribute to the increased biodeterioration in comparison with the laboratory tests.

For the initial germination of the fungal spores, a minimum amount of organic substrates is required, which can be present in natural conditions (option 3) as the surface contamination of the coatings, and in laboratory tests as trace amounts of impurities (option 1), or as components of the artificially added Czapek-Dox medium (option 2). It is known that the degree of heterotrophic CO_2_ fixation is highly dependent on the availability of easily degradable organic-carbon sources. The fungi fixed relatively more CO_2_ at lower organic carbon concentrations [25]. The main portion of the fixed CO_2_ (98–99%) was found in the extracellular metabolites, while only approximately 1% CO_2_ was incorporated into the microbial cells [26]. This is probably why the biosynthesis of the organic acids by the micromycetes occurs even when their growth on the surface of samples is weak. The role of the organic acids in biodeterioration is confirmed by the fact that their most active producer, namely *A. niger* (Figure 5), was dominant among all of the studied fungi growing on the surface of the polysiloxane coatings.

Among the fungi, there are also some diazotrophs, although there is little information about them in the scientific literature. A relatively narrow taxonomic group of fungi that form endotrophic mycorrhiza, belonging to the genera *Phoma* (in heather plants) and *Rhizostonia* or *Orcheomyces* (in orchids), is capable of assimilating atmospheric nitrogen [27]. The question of nitrogen fixation by the genera *Aspergillus* and *Penicillium* is still debatable, although at the beginning of the twentieth century, nitrogen fixation by *Aspergillus niger* and *Penicillium glaucum* was recorded [28]. We did not consider this issue in detail, since the growth of the studied micromycetes could also be supported by trace amounts of nitrogen-containing pollution which are presented in the atmosphere [29].

## 5. Conclusions

Summarizing the obtained results, we made the following conclusions: (i) the main contribution to the damage of polysiloxane coatings is made by micromycetes; (ii) micromycetes of the genera *Aspergillus, Penicillium*, *Fusarium*, and *Alternaria* are capable of limited growth on the surface of polysiloxane coatings; (iii) dominant fungi of the genera *Aspergillus* and *Penicillium* are producers of organic acids, among which a high level of oxalic acid was registered for all of the strains; (iv) the results of laboratory tests correlate with the tests in the natural conditions of tropics; (v) the created polysiloxane coating is promising for use in practice, since it retains significant resistance to abrasion, adhesion strength to the surface of organic glasses, and transparency of all of the samples at a level of more than 90% under aggressive conditions in the tropics.

## Figures and Tables

**Figure 1 microorganisms-10-01597-f001:**
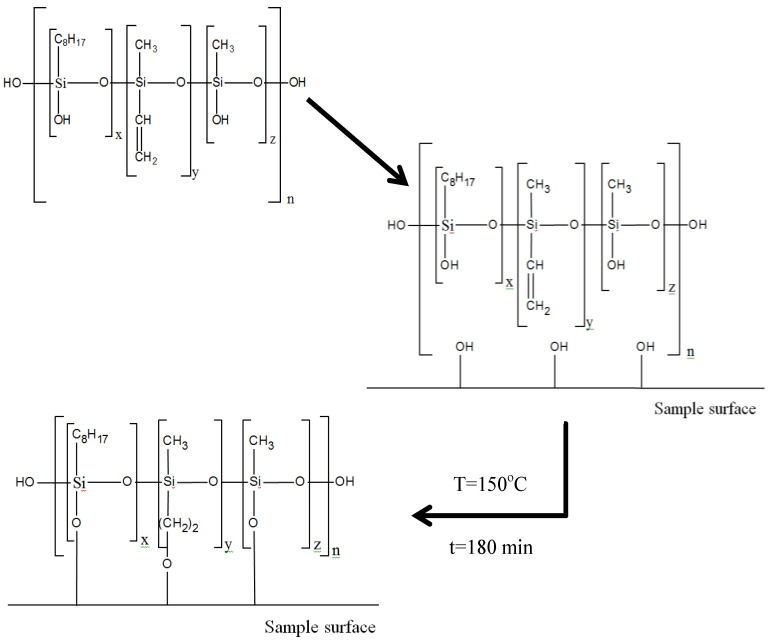
Scheme of varnish composition formation on the surface of organic glass samples. x = 0.05n, y = 0.9n, z = 0.05n (n is the number of monomers in a polymer molecule). For better visualization, the blocks in the formulas are allocated conditionally, since in reality the rotation of functional groups in molecules is chaotic.

**Figure 2 microorganisms-10-01597-f002:**
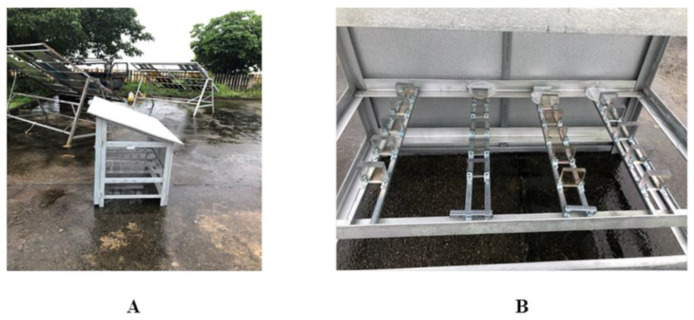
The placement of samples at the testing station, Tropical Center, Nha Trang, Vietnam. (**A**)—concrete platform with installations for placing organic glass samples; (**B**)—installation with placed samples.

**Figure 3 microorganisms-10-01597-f003:**
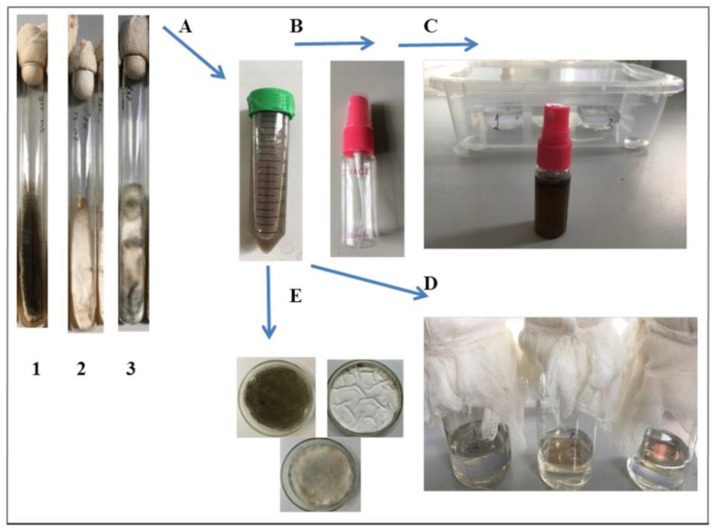
Scheme of polysiloxane coatings laboratory trials. Experimental stages: preparing spore suspension (**A**); applying suspension into sprinkler (**B**); incubating spattered samples in chambers (**C**, option 1); incubating samples in Czapek-Dox medium containing mixed spore suspension (**D**, option 2); controlling spore viability (**E**). 1—*Aspergillus niger*; 2—*Penicillium chrysogenum*; 3—*Aspergillus puulaauensis*.

**Figure 4 microorganisms-10-01597-f004:**
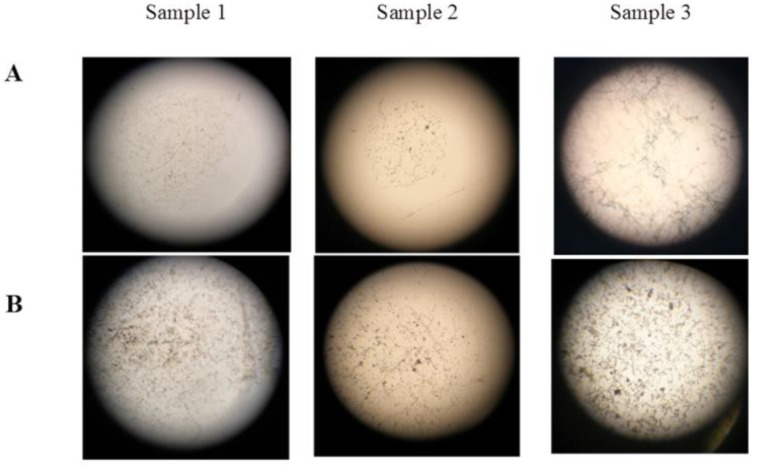
Microscopy of fouling area on polysiloxane coatings, treated according to option 1. Magnification 64× (**A**); 160× (**B**).

**Figure 5 microorganisms-10-01597-f005:**
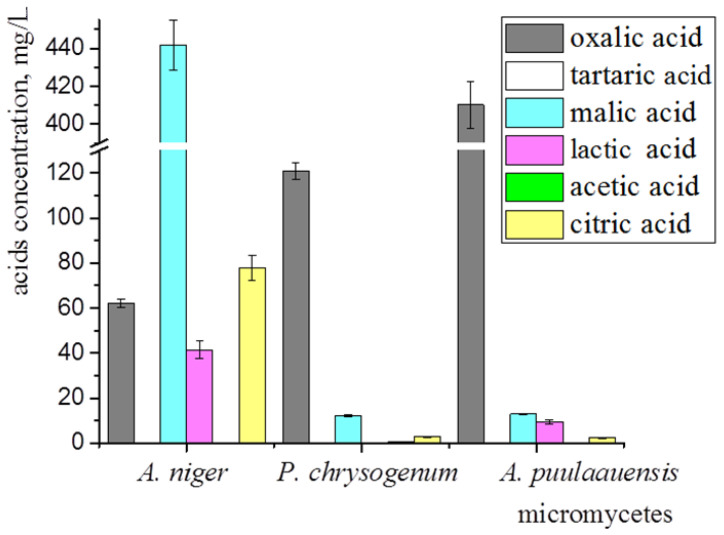
Organic acids synthesized by micromycetes growing on Czapek-Dox medium at 21st day of cultivation. Average data of three independent experiments with standard deviation are presented.

**Figure 6 microorganisms-10-01597-f006:**
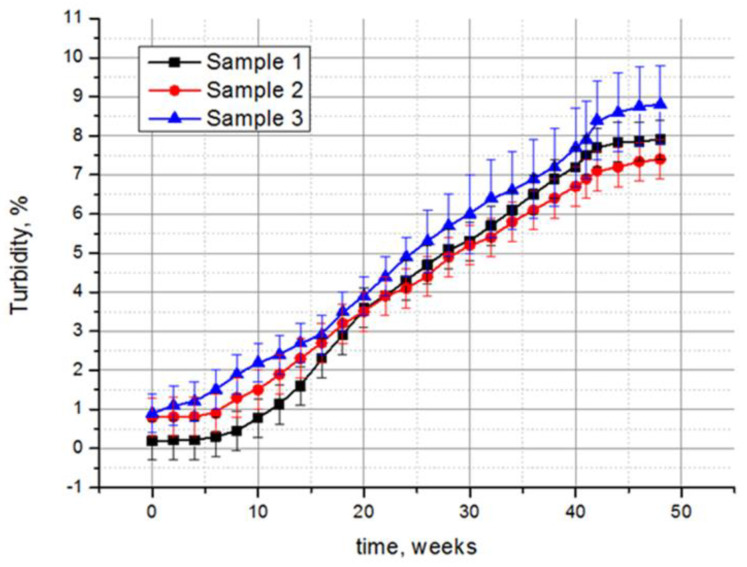
Increase in turbidity of polysiloxane-coated samples during exposure at the Tropical Centre, Vietnam. For 100% turbidity, the complete absence of optical radiation transmittance was taken. Average data of three independent experiments with standard deviation are presented.

**Figure 7 microorganisms-10-01597-f007:**
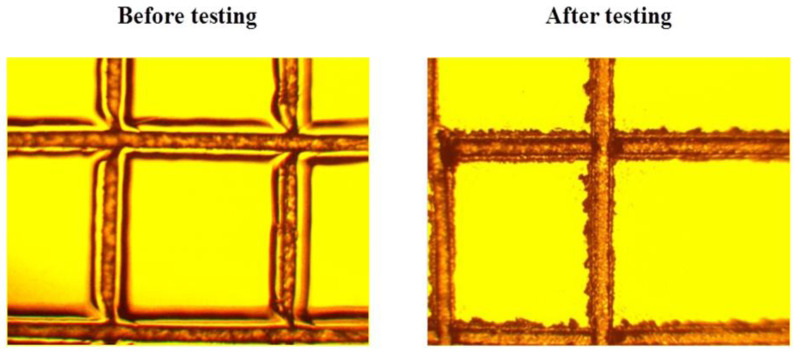
An example of slight deterioration in the adhesion strength of the coating for sample 1 treated according to option 2.

**Table 1 microorganisms-10-01597-t001:** Optical–mechanical properties of the polysiloxane-coated samples after laboratory and natural trials.

No	Sample	Trials Option *	Transmittance, (τ)	Turbidity, (H), % **	Abrasion Degree ***
1	Monolithic polycarbonate	1	0.65	2.8	−
		2	0.74	3.1	±
		3	0.53	7.9	±
		4	0.78	0.2	−
2	Monolithic polymethyl methacrylate	1	0,68	1.8	−
		2	0.67	2.1	−
		3	0.55	7.4	±
		4	0.76	0.8	−
3	Polycarbonate triplex	1	0.67	1.7	−
		2	0.66	2.9	−
		3	0.56	8.8	±
		4	0.79	0.9	−

* 1—laboratory trials without substrates; 2—laboratory trials with substrates; 3—natural trials in Vietnam; 4—non-treated control samples. ** For 100% turbidity the complete absence of optical radiation transmission was taken. *** (–)—no abrasion, (±)—slight abrasion.

## Data Availability

Not applicable.

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
