# Peer review of "Polysiloxane Coatings Biodeterioration in Nature and Laboratory"

_microorganisms, 2022, doi:10.3390/microorganisms10081597_

Round 1

Reviewer 1 Report

In this study, the authors prepared a polysiloxane coatings for polycarbonate and polymethyl methacrylate substrates for use in tropical climates and analyzed their stability under natural and laboratory conditions, focusing on the damaging effects of microbes. This study has the potential to help to understand the mechanism of biodeterrioration, as well as to identify potentially undesirable fungal communities or highly aggressive species. This study is interesting and the manuscript is well-written, thus I suggest accepting this manuscript for publication in Microorganisms with minor revisions.

Specifically,

The format of section 2.1 ‘Polysiloxane coating’ needs to be modified.

The section ‘Materials’ is missing.  

Author Response

We would like to thank the Reviewer for his detailed analysis, thoughtful comments, and efforts towards improving our manuscript. We sincerely apologize for some mistakes and inaccuracies that we have tried to correct.

The format of section 2.1 ‘Polysiloxane coating’ needs to be modified.

The section was rewritten (lines 76-98).

The section ‘Materials’ is missing.

All materials with CAS numbers are added to the section 2.

Reviewer 2 Report

This article explores the biodegradation of polysiloxane coatings applied on three different transparent polymers. The degradation behaviors of coated specimens were quantitatively explored and compared under simulated lab conditions and the actual field environments. The authors identified that micromycetes contributed the most to the damage of the coatings. Producers of various organic acids were revealed and their effects on the coating damage were evaluated. In general, this is a well designed study and the interpretation is reasonable. I recommend publishing this article in the current form.

Author Response

We are very grateful to the Reviewer for his attentive attitude to our work and his positive conclusion.

Reviewer 3 Report

The article "Polysiloxane coatings biodeterrioration in nature and laboratory" is a paper that does not meet the quality standards required for a scientific research. Some of the problems are listed below:

Use italics for the fungal strain names across the manuscript.

The English language needs some polishing for style and typos (e.g. starting with the title “biodeterrioration”!; “Applied Bisystems”; “4×106” etc; In addition, please use same font type across the manuscript – see rows 78-90). Carbon dioxide is occasionally written CO2 and not CO2.

Some phrases are obvious truisms like rows 54-57 “...nanoparticles.....even in cases where the particle sizes are smaller than wavelengths of the optical range” – a nanoparticle must be under 100 nm to be qualified as nano, which is under 400 nm (visible limit towards UV domain). Or phrase from rows 244-245 – the samples where microorganisms have nutrients will exhibit a higher growth rate and therefore of course will be more deteriorated!!!

The clarity of the method is missing. Authors should explain how the coating was made with full references, e.g. “.... in the amount of 10% of sediment weight.” – what sediment? final solid mass of the coating? How was that weighted? In addition the phrase from rows 80-81 suggest that the flask has the mass of 0.01 g.

GOST standards in English should be presented as a link in bibliography. International community cannot access and read national Russian standards, in Russian.

Source of Czapek's medium, or if made “in house” the exact composition? The Cazpek’s medium from row 113 is the same with Czapek-Dox medium from row 152?

Abbreviations must be explained at first use (e.g. PCR).

Use uniform notation for measurement units (now for litre are used both l and L like at rows 82 and 128 but also elsewhere across the manuscript). Personally, I would recommend the use of L.

In figure 1 not all the covalent bonds are straight lines as they should be.

What measurement units are used by authors in the statement “sensitivity range of 0.4 – 1.1 μM”?

The optical measurements are made with a photodiode and a light emitting diode, with no other indication about the electronics used (if any), the experimental conditions (closed box, open setup?), which in my opinion is unacceptable for a scientific article. A UV-Vis spectrophotometer is a cheap, available instrument, with known resolution, precision, accuracy etc. There is no need to improvise it.

Pressing force from row 186 can not be expressed in grams. A force is always expressed in Newton.

For cross-cut test what tool was used to do the cuts, as this influences the results?

Samples were kept for 7 months at the climatic testing stations.... Tropical Center (row 106). Analysis of samples exposed for 12 months in the Tropical Center (row 202). Figure 6 presents a time line of 48 weeks, which is not same thing with 12 months and at row 310 are mentioned 50 weeks. Authors should make up their minds about the exposure time.

“in the tropics, temperatures ranged from 26°C to 38°C with a humidity of about 81%” – lack of rigorosity in such statements. Is that an average? The tropics still have a rainy and a dry season. Web sources give an average annual temperature of 26.7 °C.

Author Response

We thank the reviewer for their careful attention to our work and detailed analysis. Thanks to this, we were able to significantly improve the text of our manuscript. We apologize for any mistakes or inaccuracies.

1. Use italics for the fungal strain names across the manuscript.

Corrected.

2. The English language needs some polishing for style and typos (e.g. starting with the title “biodeterrioration”!; “Applied Bisystems”; “4×106” etc; In addition, please use same font type across the manuscript – see rows 78-90). Carbon dioxide is occasionally written CO2 and not CO2.

Corrected.

3. Some phrases are obvious truisms like rows 54-57 “...nanoparticles.....even in cases where the particle sizes are smaller than wavelengths of the optical range” – a nanoparticle must be under 100 nm to be qualified as nano, which is under 400 nm (visible limit towards UV domain).

Corrected (line 55).

4. …Or phrase from rows 244-245 – the samples where microorganisms have nutrients will exhibit a higher growth rate and therefore of course will be more deteriorated!!!

There are lines 327-329: “As expected, in the presence of additional nutrients for fungal growth in Czapek-Dox medium (option 2) biodeterioration was more prominent compared to option 1 without added nutrients” We found this phrase important, since this confirms that the main destructive factor is the production of organic acids, and not the use of polysiloxane itself by fungi as a nutrient source.

5. The clarity of the method is missing. Authors should explain how the coating was made with full references, e.g. “.... in the amount of 10% of sediment weight.” – what sediment? final solid mass of the coating? How was that weighted? In addition the phrase from rows 80-81 suggest that the flask has the mass of 0.01 g.

Polysiloxane coating section of “Materials and Methods” was rewritten.

6. GOST standards in English should be presented as a link in bibliography. International community cannot access and read national Russian standards, in Russian.

GOST standards were eliminated, the detailed description of methods was given.

7. Source of Czapek's medium, or if made “in house” the exact composition? The Cazpek’s medium from row 113 is the same with Czapek-Dox medium from row 152?

Corrected through the text as Czapek-Dox medium.

8. Abbreviations must be explained at first use (e.g. PCR).

Corrected (line 128).

9. Use uniform notation for measurement units (now for litre are used both l and L like at rows 82 and 128 but also elsewhere across the manuscript). Personally, I would recommend the use of L.

Corrected to mL.

10. In figure 1 not all the covalent bonds are straight lines as they should be.

Corrected.

11. What measurement units are used by authors in the statement “sensitivity range of 0.4 – 1.1 μM”?

Corrected to μm (micrometer).

12. The optical measurements are made with a photodiode and a light emitting diode, with no other indication to n about the electronics used (if any), the experimental conditions (closed box, open setup?), which in my opinion is unacceptable for a scientific article. A UV-Vis spectrophotometer is a cheap, available instrument, with known resolution, precision, accuracy etc. There is no need to improvise it.

We have used the spectrophotometer of our personal construction described in (Danilaev et al., 2021). The authors of the article developed a photometer with high stability of characteristics over time to study sedimentation processes. However, the characteristics of this photometer allow it to be used to measure the transmittance and turbidity of optically transparent samples. Its scheme and the results of reference glass slides optical density measurements are given in Appendix A.

13. Pressing force from row 186 can not be expressed in grams. A force is always expressed in Newton.

Corrected to the pressing force ≈ 2 N (line 194)

14. For cross-cut test what tool was used to do the cuts, as this influences the results?

Included to the text (lines 203-205): Transverse incisions were made manually using a knife (ISO 2409:2013) with a V-shaped blade: blade thickness 0.43±0.03 mm; sharpening angle 25±3°; sharp edge width ~0.05 mm.

15. Samples were kept for 7 months at the climatic testing stations.... Tropical Center (row 106). Analysis of samples exposed for 12 months in the Tropical Center (row 202). Figure 6 presents a time line of 48 weeks, which is not same thing with 12 months and at row 310 are mentioned 50 weeks. Authors should make up their minds about the exposure time.

Corrected to 48 weeks (corresponds to 12 month) through the text.

 16. “in the tropics, temperatures ranged from 26°C to 38°C with a humidity of about 81%” – lack of rigorosity in such statements. Is that an average? The tropics still have a rainy and a dry season. Web sources give an average annual temperature of 26.7 °C.

We agree with the reviewer and indicate the average temperature of 26.7 °C (line 350).

Round 2

Reviewer 3 Report

The authors have responded to my comments and have addressed all my concerns therefore, I suggest publishing the paper in the current form.